# A Sensing-Based Visualization Method for Representing Pressure Distribution in a Multi-Zone Building by Floor

**DOI:** 10.3390/s23084116

**Published:** 2023-04-19

**Authors:** Jiajun Jing, Dong-Seok Lee, Jaewan Joe, Eui-Jong Kim, Young-Hum Cho, Jae-Hun Jo

**Affiliations:** 1Department of Architectural Engineering, Inha University, Incheon 22212, Republic of Korea; 2Department of Architectural Engineering, Keimyung University, Daegu 42601, Republic of Korea; 3Division of Architecture, Inha University, Incheon 22212, Republic of Korea; 4School of Architecture, Yeungnam University, Gyeongsan 38541, Republic of Korea

**Keywords:** pressure-sensing system, multi-zone buildings, grid, coordinate system, visualization, pressure mapping

## Abstract

Airflow in a multi-zone building can be a major cause of pollutant transfer, excessive energy consumption, and occupants discomfort. The key to monitoring airflows and mitigating related problems is to obtain a comprehensive understanding of pressure relationships within the buildings. This study proposes a visualization method for representing pressure distribution within a multi-zone building by using a novel pressure-sensing system. The system consists of a Master device and a couple of Slave devices that are connected with each other by a wireless sensor network. A 4-story office building and a 49-story residential building were installed with the system to detect pressure variations. The spatial and numerical mapping relationships of each zone were further determined through grid-forming and coordinate-establishing processes for the building floor plan. Lastly, 2D and 3D visualized pressure mappings of each floor were generated, illustrating the pressure difference and spatial relationship between adjacent zones. It is expected that the pressure mappings derived from this study will allow building operators to intuitively perceive the pressure variations and the spatial layouts of the zones. These mappings also make it possible for operators to diagnose the differences in pressure conditions between adjacent zones and plan a control scheme for the HVAC system more efficiently.

## 1. Introduction

Airflows between internal zones and between indoor and outdoor spaces can cause problems in relation to indoor environmental quality (IEQ) and building energy consumption, such as air pollutant transfer [1,2,3], excessive energy consumption due to air infiltration [4,5,6,7], as well as occupants’ discomfort [8,9]. The key to mitigating these problems is to investigate in real time the variations in indoor environmental parameters such as the concentration of air pollutants, temperature, humidity, and the absolute pressure.

The potential technologies of wireless sensor networks (WSNs) and the Internet of Things (IoT) contribute to the development of real-time monitoring systems for indoor environmental parameters. A wireless sensor network normally consists of a large number of wireless sensing devices to collect data from working environments, so that processes may be centralized to databases, and the environmental parameters of space of interest may be monitored remotely [10,11,12]. The IoT allows the implementation of a WSN that connects all objects to the cloud server services with automatic and timely transmissions, data processing, data analysis, and visualization [13,14]. Intelligent monitoring and control systems using the IoT have been developed and commercialized in smart buildings for the purposes of monitoring indoor environmental quality (IEQ), evaluating occupant comfort, and predicting energy consumption [15,16].

From among the monitored parameters, the IEQ is arguably the most important, since the IEQ can reflect unexpected variations in indoor environmental conditions and provide signals for the control of HVAC systems [17]. The IEQ can be described as a combination of indoor thermal parameters, indoor air pollutant parameters, as well as lighting, acoustic, and spatial factors [18,19]. Among these parameters, indoor thermal parameters [20] and air quality-related parameters [21] are typically applied to monitor the variation of IEQ inside buildings. In particular, comprehensive IEQ monitor systems have recently been developed for inspecting variations in multiple indoor environmental parameters. For example, Parkinson et al. [10] developed an indicative wireless IEQ sensor network called SAMBA to automatically provide timely and actionable IEQ data to building operators.

However, variations in these IEQ parameters are mainly influenced by airflows between adjacent zones and across building envelopes [22,23,24]. It is thus particularly critical to monitor airflows for an adequate evaluation of indoor environmental parameters. Moreover, airflows inside a building are rarely measured, and real-time monitoring processes are also virtually nonexistent [25]. Airflow occurs due to pressure differences across the building components and envelopes, and measurements of pressure differences between adjacent zones or of absolute pressure values in the respective zones provide indications of the airflow characteristics within and around a building. In addition, the direction of airflows between two adjacent zones in a multi-zone building varies with the direction of pressure difference, and airflow always infiltrates from the zone with high pressure to the zone with low pressure. The real-time pressure monitoring allows for the convenient detection of variations in airflow direction between adjacent zones.

Recent advances in technology have made real-time measuring of pressure variations practical, low-cost, and widely available. Furthermore, as the availability of real-time data provided by pressure sensors in a building has increased significantly, another topic warranting consideration is the visualization of the monitored pressures, because it is essential for occupants and building operators to understand and manage IEQ parameters with intuitive, visualized results. Protopsaltis et al. [26] reviewed the tools available for IoT data visualization, and visualized charts were divided into various types for specific purposes such as scatterplots, histograms, bar plots, bubble charts, boxplots, and line charts. Maps, timelines, diagrams, and graphs are also extensions of charts used in some special cases [26]. Conventionally, line charts are the most frequently used visualization tool for representing indoor environmental parameters [27]. Niedrite et al. [28] classified line charts used in the IAQ domain into eight types based on various intents of visualization. However, line charts are ineffective in both displaying real-time information for many sensors and identifying relative differences in spatial and geographical data. Therefore, as an alternative, mappings are used to visualize changes in monitored data over time and space, as in the case of concentration mapping for urban air pollutants [29,30,31]. This visualization method applied in the building environment domain is conducted to represent spatiotemporal variations in indoor thermal parameters, such as PM concentration [32], IEQ distribution [33], spatial temperature distribution [34], relative humidity [35] and so on.

Despite progress in depicting spatiotemporal variations of IEQ parameters using mappings, some limitations remain that need to be improved. First, these visualized mappings are only for a large space or buildings with a single zone rather than multiple segmented zones. There are no related studies and methods for visualizing the spatiotemporal distribution of environmental parameters in a multi-zone building. Second, only thermal environmental parameters are visualized by 2D or 3D spatial mappings. The characteristics of pressure difference between adjacent zones, which are closely related to the changes in thermal environmental parameters, are not monitored or visualized.

Therefore, the aim of this study is to present an intuitive visualization method that allows for the explanation of absolute pressure distribution on the target floor using the pressure mappings. The proposed method makes it possible to clarify the characteristics of real-time pressure difference variations between adjacent zones. To verify the feasibility of this method, field pressure measurements are performed in actual low-rise and high-rise multi-zone buildings located in South Korea. The pressure-sensing system was deployed in all measurement zones to monitor pressure variations in real time. Then, the visualized pressure distribution mappings were generated through grid formation and coordinate establishment processes for the building floor plans. The pressure distribution mappings by floor, as intuitively recognizable results of the visualization for pressure conditions, not only enable building operators and occupants to fully understand the pressure characteristics within the measurement zones, but also provide a basis for the control of IEQ factors and the operation of HVAC systems.

The structure of this study is organized as follows. Section 2 presents the methodology of this visualization process for monitored pressure mappings. Section 3 induces the developed pressure-sensing system for pressure monitoring including the overall architecture and the detailed configurations of applied devices. In Section 4, characteristics of target buildings and pressure variations of each zone are explained, and then the pressure mappings for target buildings are determined using the proposed visualization method. Finally, Section 5 concludes the main work of this study with a detailed consideration for future studies.

## 2. Methodology

Indoor environmental parameters are typically influenced by internal airflows. However, existing methods make it difficult to investigate airflows directly. Even though various monitoring systems based on WSN and IoT techniques are frequently employed within buildings to monitor specific environmental parameters, it needs to be noted that internal airflows are induced by pressure differences across the building components. As such, this study proposes a method for displaying the pressure distribution within a multi-zone building by floor. As shown in Figure 1, this method is carried out by the following four steps.

Step I: Grid formation based on the floor plan of the target building.

Based on the building floor plan, the numbers of columns along two sides of the target building are first identified. (In the target building shown in Figure 1, there are eight columns and six columns along the two directions).T building floor plan can first be divided into groups of small rectangles based on the locations of the columns. (The numbers of rectangles for the target floor in Figure 1 are 30 inside the boundary of the building and 40 outside).The number of measurement zones is identified, and the maximum number of zones between two adjacent columns is selected. (There are nine measurement zones for the target floor, and the maximum number of zones between any two columns is set to two).Following the preliminary grid layout and the maximum number of zones between two adjacent columns, the final grid of the building floor plan is obtained. (The total numbers of grid squares for the target building floor in Figure 1 are 105 inside the boundary of the building and 147 outside).

Step II: Establishment of a coordinate system for the measurement zones.

A Cartesian coordinate system is established, with the *X*-axis and *Y*-axis oriented along the two selected sides of the target building. The location of each grid square derived in Step I is expressed as a single coordinate point.The measurement zones are represented by various groups of coordinates (*x*, *y*).The coordinate system of measurement zones can be obtained for each floor through the same procedures.

Step III: Determination of spatial and numerical mapping relationships.

The coordinates of the measurement zones are organized by floor.Spatial and numerical mapping relationships (*x*, *y*, *z*) are determined using the collection of coordinates and the monitored pressure values of each zone.

Step IV: Drawing of the pressure distribution mappings by floor.

Based on the spatial and numerical mapping relationships of the measurement zones, the pressure distribution mappings are drawn using the three-dimensional plotting tool in MATLAB.

As noted in the detailed explanation above, the pressure distribution mappings of each floor can be developed based on the monitoring of pressure values in each zone.

## 3. Development of the Pressure-Sensing System for Pressure Monitoring

A pressure-sensing system for pressure monitoring was developed by the authors for this study, where the system utilizes real-time wireless network transmission to store the monitored absolute pressure values taken from the measurement zones. This system includes multiple Slave devices for data monitoring and a Master device acting as a data receiver and processor. 

Figure 2 illustrates the overall architecture of the developed pressure-sensing system. Before the actual measurement work, the stational position of devices on the floor surface should be checked by field investigation to ensure that each position will not affect the normal work and life of the occupants. Unlike temperature and humidity, the absolute pressure on the floor surface always remains consistent everywhere in a single zone without partition, and there is no apparent pressure difference between different points on the floor surface. Then, the limited number of Slave devices are connected to a Master by a wireless network. These Slave devices are calibrated through the multi-communication module of the connected Master device to set the system time, data acquisition interval, standard pressure, and other parameters. Temperature, humidity, and absolute pressure are sensed by the installed several Slave devices and are further collected, retrieved, and processed by the wireless-connected Master device to generate preliminary visualized results. In addition, building operators can also perform detailed data processing manually after collecting data from the Slave devices, so as to control building systems more efficiently in accordance with the analyzed results.

Figure 3 shows the detailed configurations of the Master and Slave devices. As shown in Figure 3a, the Master device is mainly composed of five modules: (1) the Setup module, (2) Storage module, (3) Display and data processing module, (4) Multi-communication module, and (5) Power module. In particular, the multi-communication module transmits a signal for setting up the parameters to the Slave devices and receives the monitored data via the wireless network. The monitored raw data received from the group of Slave devices are further processed and visualized by the Display and data processing module. 

Figure 3b depicts the configurations of the Slave devices. There are mainly six modules in a Slave device, including the Setup module, Sensor module, Display module, Data collection Module, Multi-communication module, and Power module. Before the formal measurement, offsets in system time, temperature, and humidity need to be adjusted through the Multi-communication and Setup modules. The sensor module monitors the indoor environmental parameters of absolute pressure, temperature, and humidity with sensor 1 and sensor 2 and records the sets of raw data. Table 1 shows the specifications of sensor 1 used in the Slave devices. Each Slave device monitors these parameters at fixed intervals, and the accumulated data are stored as sets of raw data by the Data collection module. Real-time variations of the parameters can be directly shown by the Display module. Thus, each Slave device can also serve as a pressure sensor without being connected to the Master device and can record data in real time. In addition, the multi-communication module can transmit the collected data to the Master device to express the raw data as visualized line charts.

## 4. Pressure Mappings for Target Buildings Based on the Proposed Visualization Method

### 4.1. Description of the Target Building

In this study, two multi-zone buildings are selected as target buildings for the real-time monitoring of variations in absolute pressure. Target building A is a low-rise office building of four stories located in Incheon, Korea. Target building B is a high-rise residential building with 49 floors above ground and 5 floors underground located in Seoul, Korea. General information regarding the two target buildings is shown in Table 2.

The number and layout of measurement zones in each target building were identified from field investigations. Figure 4a shows the detailed distribution of monitored zones and the locations of Slave devices on each floor. A total of 22 zones are marked with various colors in target building A: 4 zones on the 1st floor, 5 zones on the 2nd floor, 4 zones on the 3rd floor, and 9 zones on the 4th floor. These zones have different functions, such as offices, meeting rooms, and warehouses. In addition to the Slave devices deployed in these 22 zones, Slave devices were also deployed outside on the 1st floor and the roof. The outside pressure values for the 2nd floor, 3rd floor, and 4th floor was estimated by monitored outside pressure on the 1st floor and the roof.

Figure 4b shows the floor plan and layout of measurement zones for target building B. The typical floor plan consists of a central core surrounded by three elevator shafts and a stair shaft, an elevator hall, and three residential units. Not all floors in target building B had Slave devices installed. As shown in Figure 4b, the 1st floor, the 10th floor, the 20th floor, the 30th floor, the 40th floor, and the 47th floor were selected as representative floors for monitoring pressure variations. For the 10th floor, the 30th floor, and the 40th floor, Slave devices were deployed only in the residential unit 2 (U2), the Elevator Hall (EV.H), and the outside, due to the circumstances of the building not allowing the installing of devices within the elevator shafts on these floors. One more device was installed in the elevator shaft on each of the 1st and 47th floors. No device was placed outside on the 20th floor, and the pressure on the 1st and PIT floors were used to estimate the outside pressure value on the 20th floor. The pressure values within the elevator shafts on the 10th floor, the 20th floor, the 30th floor, and the 40th floor were also estimated using the data for the elevator shafts on the 1st and 47th floors. Slave devices were also placed in the lobby and vestibule on the 1st floor due to its unique layout.

### 4.2. Pressure Variation during Measurement Periods

As a part of this study, pressure measurements were taken for the two target buildings in different seasons. A 24 h measurement was conducted in building A on 3 September 2022, and a 6 h measurement was conducted in building B on 29 December 2020. To minimize the effect of other factors, such as elevator use by occupants, the opening of doors, and external winds, on absolute pressure variation, field measurements were performed from 0 a.m. to 24 p.m. on Saturday for building A and from 0 a.m. to 6 a.m. for building B. During the measurement period, elevators and other ventilation devices were not operated. The values of absolute pressure were monitored every 30 s. Due to the circumstances of the target buildings, the outside pressure and elevator shaft pressure for some floors were calculated using the equation given below.
(1)$$P_{ith}=P_{1st}+\rho gH$$

Here, $P_{ith}$ is absolute pressure either outside the building or inside the elevator shaft on the *i*-th floor in Pa, $P_{1st}$ is the absolute pressure either outside the building or inside the elevator shaft on the 1st floor in Pa, ρ is the inside or outside air density in kg/m^3^, g is the gravitational acceleration in m/s^2^, and H. is the absolute height on the *i*-th floor.

This study also made some assumptions for the pressure variation inside the target buildings. The wind effect was not considered during the measurement period; thus, the pressure outside each side of a target building was regarded as the same for each floor. Moreover, for the building B, it was assumed that residential unit 1, unit 2, and unit 3 have the same absolute pressure at any given moment. We also assumed that the pressure variations inside all elevator shafts on the same floor of building B are also the same. In particular, the absolute pressure values were monitored for the essential zones only, and as the pressure variations inside the PS rooms and EPS/TPS rooms are not considered in this study, the absolute pressure values inside these rooms were assumed to be the same as those of the EV.H. 

Figure 5 shows the pressure variation of each zone of different floors taken in real time during the measurement period for target building A. Since building A is a low-rise multi-zone building, indoor pressure variation is not easily influenced by the stack effect, and there is no huge pressure difference between adjacent zones. Figure 5 indicates that the zones on different floors exhibit similar trends of pressure variation. However, it is difficult to intuitively identify changes in pressure difference between zones, especially when there are low pressure differences acting on the building components. Furthermore, the spatial relationship between zones is also neglected, and the pressure variation of a single zone only can be identified. The pressure difference between adjacent zones is more essential since airflow movement through adjacent zones in a building is caused by the pressure difference. Thus, existing folding line charts cannot provide an adequate reference for the control of pressure variation within a low-rise building.

Figure 6 shows real-time pressure variations in zones of the 1st, 10th, 20th, 40th, and 47th floor of target building B. Significant pressure differences between zones can be easily observed on the upper floors, and the directions of pressure differences can also be clearly identified, as can be seen on the 30th, 40th, and 47th floors. Moreover, the direction of pressure difference between the elevator shaft and the outside of the building remains unchanged on these floors, because of the simple layout of the building zones and the strong stack effect in this high-rise residential building. However, the magnitude of pressure difference cannot be determined from the folding line charts. In particular, for the floors close to the neutral plane of this building such as the 10th floor, the pressure differences between zones were not great and the directions of pressure differences also changed with time. 

All in all, the existing method for visualizing pressure variations with folding line charts has shortcomings for both low-rise and high-rise multi-zone buildings and cannot provide meaningful visualized results for aiding the control of pressure in each zone. For example, changes in the directions and magnitudes of pressure differences are not readily perceivable due to the low pressure differences between adjacent zones in low-rise buildings. Additionally, the information of spatial relationships cannot be identified from the line charts. For high-rise residential buildings, the line charts are ineffective in displaying real-time variations in the magnitudes of pressure differences between zones, especially on the floors close to the median neutral plane.

### 4.3. Pressure Mappings of the Target Buildings

#### 4.3.1. Grid Formation for the Floor Plan of the Target Buildings

As shown in the Figure 1, the first step of this method is to form grids for the building floor plans. Based on the number of columns along two major sides of target buildings A and B, preliminary grids can be formed for the building floor plans as shown in Figure 7 and Figure 8.

In target building A, the 4th floor has the greatest number of zones from among the four floors, and as such, the 4th floor was selected as the basis for forming a grid for the floor plan. As shown in Figure 7(a-1), the numbers of columns along the two sides are 6 and 8, and a grid for the floor plan can be created that has a total of 30 grid rectangles within the boundary of the building. It is also needed to consider the outside of the building, and the outside of the building can be represented by a total of 40 grid rectangles shown in Figure 7(a-2). However, the grid rectangles for the inside of the building thus derived cannot represent all measurement zones. Based on the condition that the maximum number of measurement zones between two columns is two, additional grid squares are added between adjacent columns, and the final grid is as shown in Figure 7(a-3). The total number of grid squares representing the internal zones of the building is 105, while the number of grid squares representing the outside of the building is 147. Then, each measurement zone inside the target building can be regarded as a group of grid squares, and the grid formed for each floor is as shown in Figure 7b–e.

For target building B, the 2nd to 27th floors and 30th to 47th floors have the same floor plan, with the 28th floor as a shelter floor and the 29th floor as a non-residential floor. Therefore, the typical floor plan is selected as the standard floor for the grid formation process for target building B. Using the information on the number of columns along the two major sides of the target building shown in the Figure 8(a-1), a preliminary grid was formulated for the floor plan of the typical floor that has 13 grid rectangles on the inside and 36 grid rectangles on the outside. Figure 8(a-2) shows that there is only one grid rectangle between two columns, meaning that the preliminary grid cannot represent all zones on this target floor. For example, the elevator shafts and the elevator hall cannot be represented with the preliminary grid. Considering that there are PS, EV.H, EV.S, and a staircase between two columns shown in Figure 8a, the minimum number of grid squares between any two columns should be four. The final grid for the typical floor plan was thus generated with 68 grid squares inside and 134 grid squares outside shown in Figure 8(a-3). The zones on the typical floor and the lobby are represented as various groups of grid squares as shown in Figure 8b,c.

#### 4.3.2. Establishment of Coordinate Systems for the Measurement Zones

The measurement zones need to be further specified based on the formulated grids of the two target buildings. A Cartesian coordinate system in two dimensions was established for each target floor, and only positive half-axes are applied to specify the measurement zones. In the coordinate system, the *X*- and *Y*-axes denote the two sides of the target buildings, and each coordinate point designated as (*x*, *y*) is considered as a representation of a grid square. Finally, various groups of coordinates can be used to represent the measurement zones of each target building based on the grid formation results. Figure 9a depicts a coordinate system representing the measurement zones on the 4th floor of target building A, where the points indicate the coordinates of the grid squares derived in the last step. Taking Z6 and Z9 as examples, the grid squares of the two zones can be replaced by two groups of coordinates as marked in Figure 9a. Figure 9b shows the coordinate system for the typical floor in target building B.

According to the coordinate systems, the groups of grid squares can be substituted by coordinate points, and the measurement zones can be described as various sets of coordinates. Taking Z6 and Z9 on the 4th floor for target building A and U2 and EV.S 2 on the typical floor for target building B shown in Figure 9 as examples, the coordinates of these measurement zones are summarized in Table 3. The same procedures are then taken for other floors. The coordinate systems of the measurement zones on each target floor can be finally established.

#### 4.3.3. Determination of Spatial and Numerical Mapping Relationship

As shown in Figure 1, before drawing the pressure distribution mappings, the spatial and numerical mapping relationships should be determined. Pressure mappings are derived from real-time pressure variations and the coordinate sets of each measurement zone. Each coordinate is then matched with its corresponding pressure value to derive the real-time pressure mapping of the target floor.

In order to display the variations in pressure at different times, the measurement period for target building A is divided into periods from AM 0:00:00 to AM 11:59:30 and from PM 12:00:00 to PM 23:59:30. For target building B, measurement periods from AM 0:00:00 to 2:59:30 and from AM 3:00:00 to AM 5:59:30 are applied for visualizing pressure distribution. Here, the mean pressure value during each measurement period is used instead of pressure data at a certain time point to avoid device errors. Table 4 shows the mean pressure of each measurement zone for different time periods. For actual use, the real-time pressure variation in each zone can be used to extract real-time pressure mappings. The setting of time intervals and start time is needed to be the same for the sensors within all the measurement zones. The continuously varying pressure mappings can be then generated with the simultaneous pressure value in each zone. 

Then, using the derived pressure values, the spatial and numerical mapping relationships (*x*, *y*, *z*) between each pressure value z and the coordinate point (*x*, *y*) are determined. Table 5 shows the examples of the spatial and numerical mapping relationships for Z6 and Z9 on the 4th floor of target building A as well as U2 and EV.S 2 on the 20th floor of target building B during two measurement periods. Finally, all the measurement zones for the target buildings are represented using the mapping relationships (*x*, *y*, *z*).

#### 4.3.4. Pressure Mapping for the Target Buildings

Based on the established spatial and numerical mapping relationship (*x*, *y*, *z*) of each measurement zone, the three-dimensional plotting tool in MATLAB is used to visualize the pressure distribution and variation of the target buildings. 

In Figure 10, pressure mappings in 2-dimensional (2D) and 3-dimensional (3D) dimensions are shown for target building A during the two measurement periods, arranged by floor. In contrast to the folding line charts shown in Figure 5, the pressure mappings combine complex spatial information with real-time pressure values for each measurement zone inside each building, illustrating the pressure difference and spatial relationships between adjacent zones.

Building operators can intuitively understand the information on the number of zones inside a building and the layout of each zone. For example, there are nine measurement zones on the 4th floor, and Z1 is adjacent to Z2, Z4, Z5, Z6, and Z7. Furthermore, there is always a lower pressure difference between adjacent zones in a low-rise building due to the weak impact of the stack effect. The pressure mappings use colors to provide a clear distinction of similar pressure values in different zones, allowing the building operator to perceive a lower pressure difference between certain adjacent zones, such as zones Z2 and Z4 on the 1st floor, Z1 and Z5 on the 2nd floor, and Z1 and Z4 on the 3rd floor.

For pressure and airflow control in a low-rise multi-zone building, it must be considered that the real-time variation of the pressure difference between adjacent zones is more important than focusing only on the real-time pressure variation of any single zone. Pressure mappings make it possible to detect real-time changes in pressure difference based on color differences. For example, on the 1st floor, it can be readily seen from the same color hues that the pressure differences between internal zones did not change significantly over the two measurement periods, even as the pressure difference between the inside and outside of the building increased in the second measurement period. In comparison to other zones, the difference between Z8 and Z9 and between the outside and inside of the building decreased on the 4th floor. 

Moreover, if the real-time pressure mappings are derived for each floor, the pressure variation characteristics can be identified by comparing these mappings. For example, the pressure of Z1 on the 1st floor was always the highest among all the measurement zones, while the pressure of Z3 was always the lowest. Such pressure characteristics can also be found on other floors. Thus, the pressure mappings derived for a low-rise building provide insight both on the features of real-time pressure variation in individual zones and on the changes in pressure differences between adjacent zones.

Figure 11 shows 2-dimensional (2D) and 3-dimensional (3D) pressure mappings for target building B from AM 0:00:00 to AM 2:59:30 and from AM 3:00:00 to AM 5:59:30. Although there are not many measurement zones in target building B, the number and layout of measurement zones on each floor are clearly identified using 2D pressure mappings similar to those of target building A. Moreover, the variation of colors provides an intuitive indication of pressure distribution within each zone.

As a result of the strong stack effect, there is always a large pressure difference acting on the building components above and below the neutral plane level of high-rise buildings, which is clearly observable in the pressure mappings for the 20th, 30th, 40th, and 47th floors. A huge pressure difference is observed between EV.H and U1, U2, and U3, while the pressure difference between EV.S and EV.H is considerably smaller than the difference between EV.H and the houses. However, with reference to Figure 6, it is difficult to identify these special characteristics using line charts. 

The pressure mappings also show the characteristics of real-time pressure difference variations between adjacent zones in target building B, where the pressure differences between adjacent zones changed little over time. Moreover, except for the 10th floor, there were no apparent variations in the pressure difference between adjacent zones, as the color of each zone remained the same during both measurement periods. This indicates that the pressure difference between adjacent zones in high-rise buildings generally remains stable, even though the absolute pressure in each zone may appear to change with time.

## 5. Discussion

The derived pressure mappings for multi-zone buildings provide detection both on the visualization of pressure conditions in an individual zone and on a pressure difference between adjacent zones using color differences. Based on the mappings, variations in airflow direction across air leakage paths between adjacent zones are then clearly identified. For building A, airflows between Z4 and Z2 on the 1st floor infiltrate from Z4 with high pressure to Z2 with low pressure during the two periods. For Z8 and Z9 on the 4th floor, the airflow direction is from Z9 to Z8 in the first period, and there are no apparent airflows between these two zones in the second period due to similar colors shown in the pressure mappings. Pressure mappings for target building B also provide a clear clarification of airflow moving paths, especially on high floors with a strong stack effect. For the 20th, 30th, 40th, and 47th floors, the airflows flow outside of the building through the horizontal zones, including the elevator shaft, elevator hall, and residential units, which further results in the difference in energy consumption within various zones due to the different temperatures and changed air infiltration rates. Moreover, if there are sources of air pollutants within certain zones, the direction of air pollutant transport can be also identified using real-time pressure mappings.

All in all, the pressure mappings allow building managers to easily understand the pressure condition in each measurement zone and quickly diagnose real-time variations in airflow direction between adjacent zones with color variations. In addition, the operation of the HVAC system can also be adjusted efficiently with the reference of diagnosed results for the airflow distribution.

## 6. Conclusions

This study develops a pressure-sensing system for real-time pressure monitoring. The developed system consists of a Master device and a number of Slave devices. Field pressure measurements were conducted in actual low-rise and high-rise multi-zone buildings with the new pressure-sensing system. The system enables the real-time monitoring of environmental parameters, especially for pressure variations within zones, and provides the preliminary visualized results of folding line charts. A novel spatiotemporal visualization method was further developed to derive pressure distribution mappings using the data collected from the developed system, which is comprised of four steps. According to the information of columns and the layout of measurement zones for two target buildings, measurement zones were first represented in a grid as groups of squares. A Cartesian coordinate system was then established to represent the measurement zones as various sets of coordinates. Each coordinate was matched with a monitored pressure value to deter’mine the spatial and numerical mapping relationships. Lastly, the pressure conditions within all measurement zones for the two target buildings were visualized for each floor as pressure mappings. 

These mappings are especially suitable for visualizing the spatiotemporal distribution of pressure in multi-zone buildings considering the layout of measurement zones. Moreover, the pressure distribution mappings make it possible to detect the pressure variation characteristics of both low-rise and high-rise multi-zone buildings. For low-rise multi-zone buildings, the mappings provide a clear distinction of the pressure differences between adjacent zones by color differences. For high-rise buildings, the mappings not only reveal large pressure differences between EV.H and U1, U2, and U3, but also a small pressure difference between EV.H and EV.S, and the pressure difference always remains consistent with time.

Despite its contributions, this study also has the following limitations: first, the shapes of the two multi-zone buildings are regular, and future work will be conducted to extend the potential application of this method to multi-zone buildings having irregular shapes. In addition, the visualization procedure is not integrated with the self-developed sensing system. The display and data processing modules will be upgraded with the function of transforming the input pressure data directly into visual pressure mappings. The developed system and its related pressure mappings are also expected to interact with the HVAC system to allow a more effective management of environmental parameters inside buildings.

## Figures and Tables

**Figure 1 sensors-23-04116-f001:**
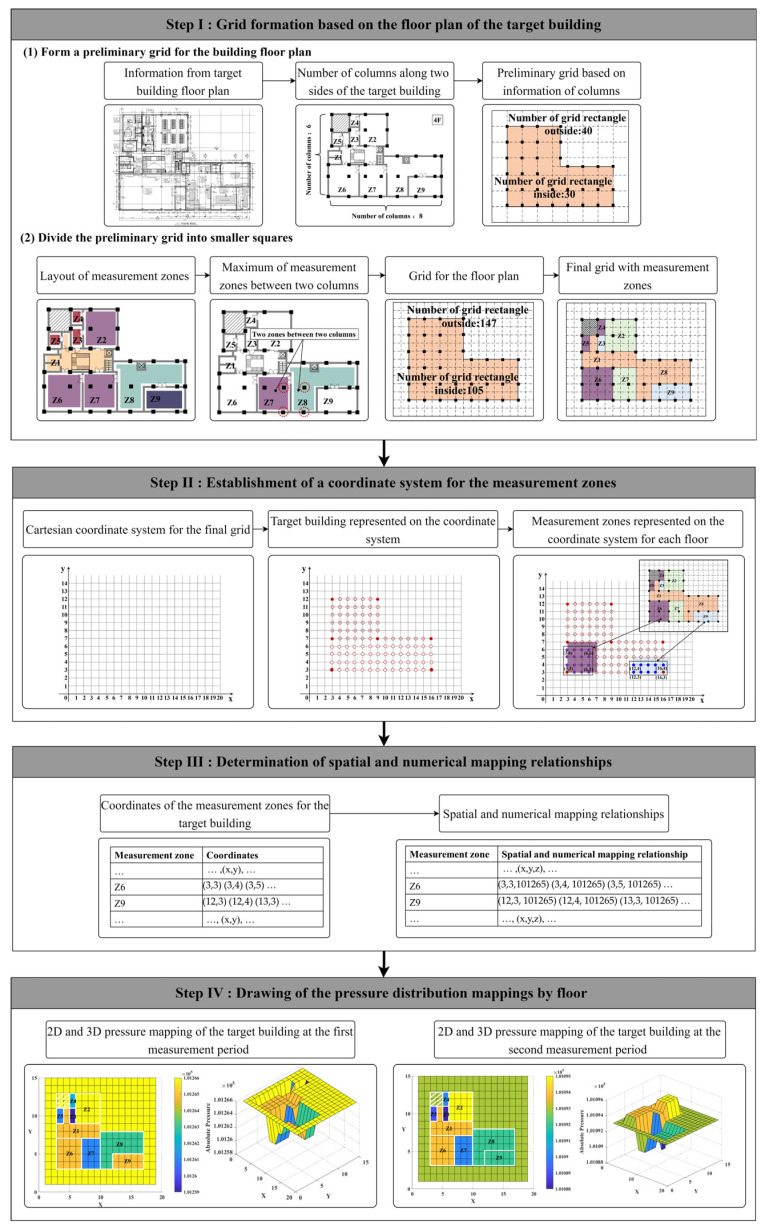
Flowchart of the proposed method.

**Figure 2 sensors-23-04116-f002:**
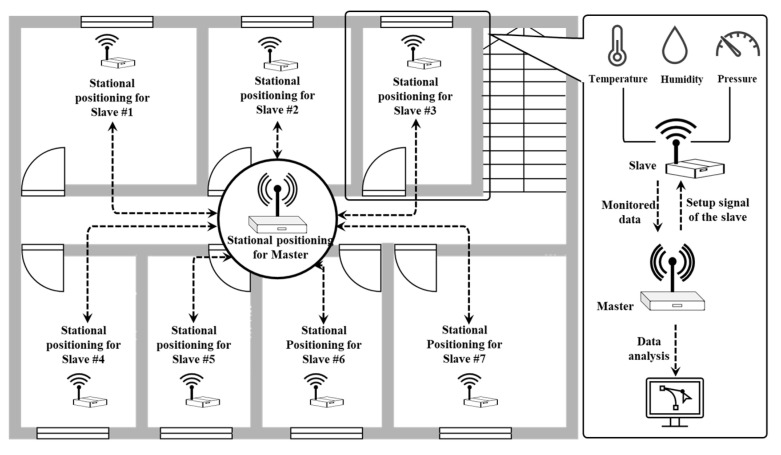
Overview of the architecture and components of the developed pressure-sensing system for pressure monitoring.

**Figure 3 sensors-23-04116-f003:**
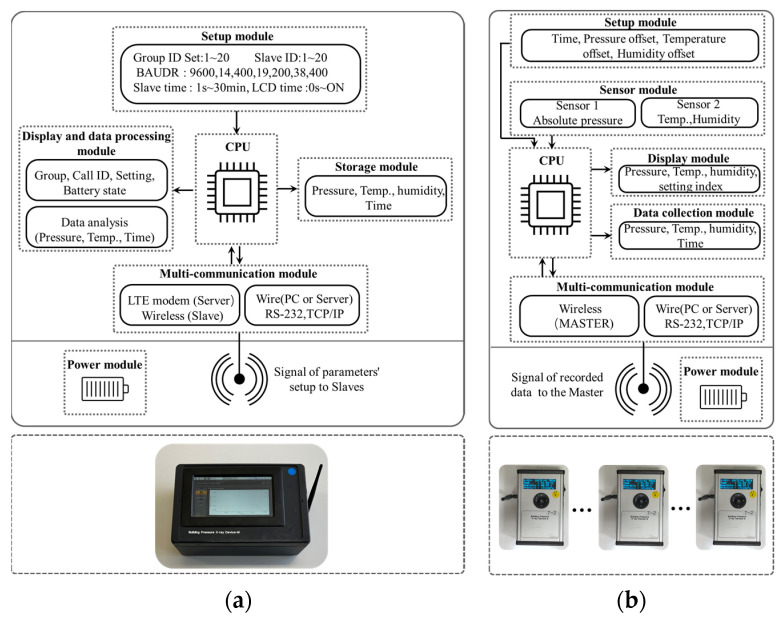
Overview of the configuration for the developed pressure-sensing system. (**a**) Configurations of the Master. (**b**) Configurations of the Slave.

**Figure 4 sensors-23-04116-f004:**
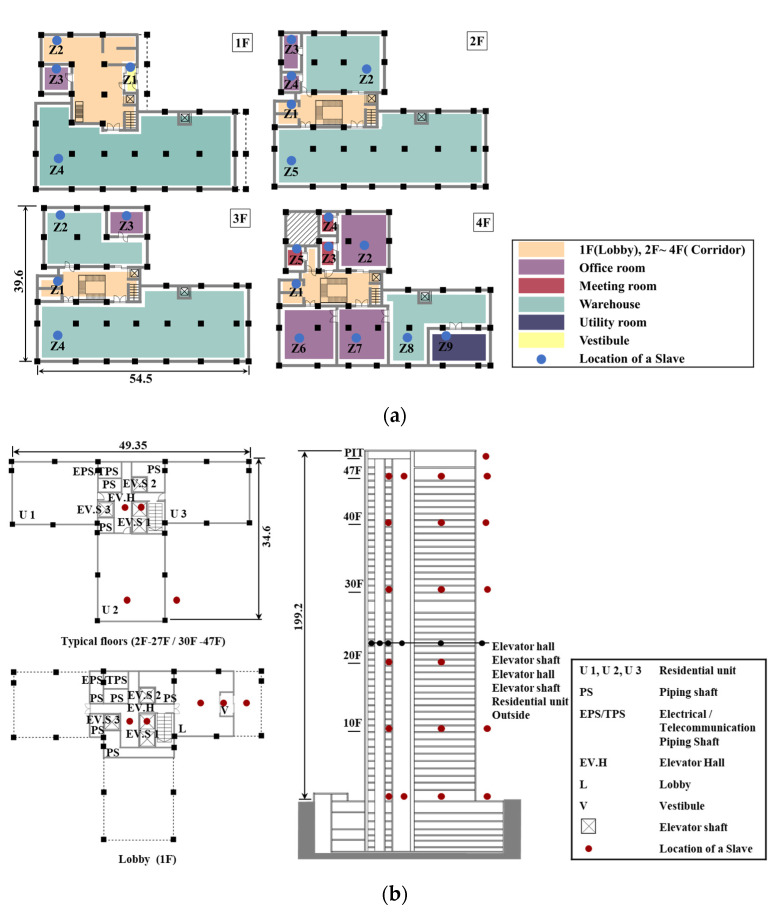
Floor plans and locations of devices for the target buildings. (**a**) Floor plan, layout of zones, and locations of Slave devices for target building A. (**b**) Floor plan, layout of zones, and location of Slave devices for target building B.

**Figure 5 sensors-23-04116-f005:**
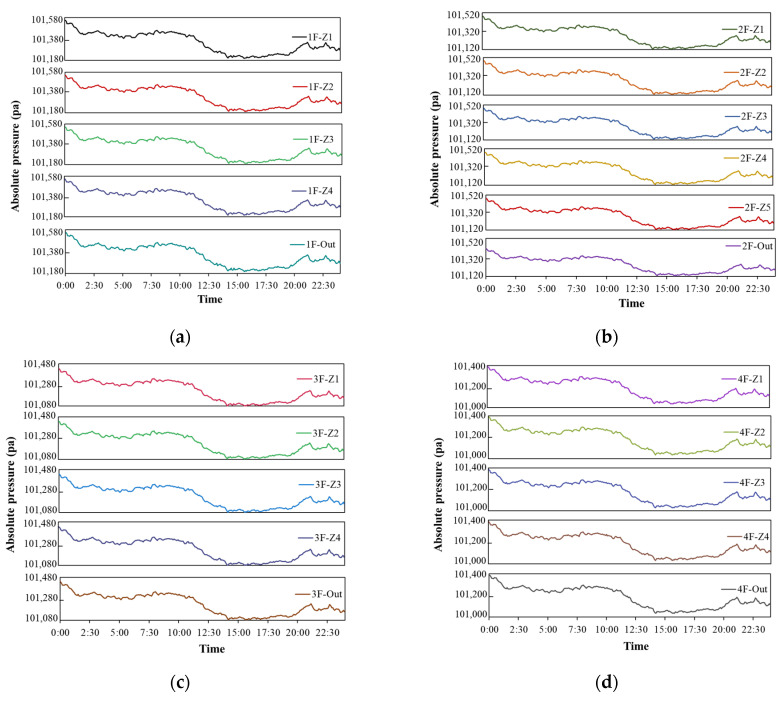
Real-time pressure variation of measurement zones for target building A. (**a**) Measured pressure variation on 1st floor. (**b**) Measured pressure variation on 2nd floor. (**c**) Measured pressure variation on 3rd floor. (**d**) Measured pressure variation on 4th floor. (**e**) Measured pressure variation on 4th floor.

**Figure 6 sensors-23-04116-f006:**
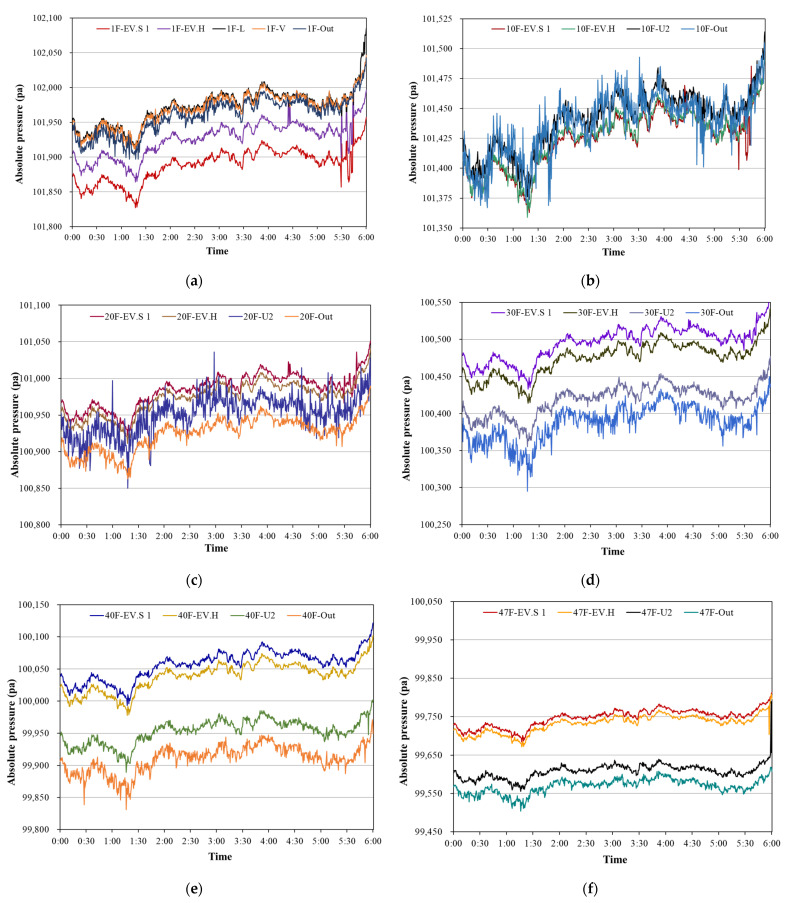
Real-time pressure variations in measurement zones for target building B. (**a**) Measured pressure variation on 1st floor. (**b**) Measured pressure variation on 10th floor. (**c**) Measured pressure variation on 20th floor. (**d**) Measured pressure variation on 30th floor. (**e**) Measured pressure variation on 40th floor. (**f**) Measured pressure variation on 47th floor.

**Figure 7 sensors-23-04116-f007:**
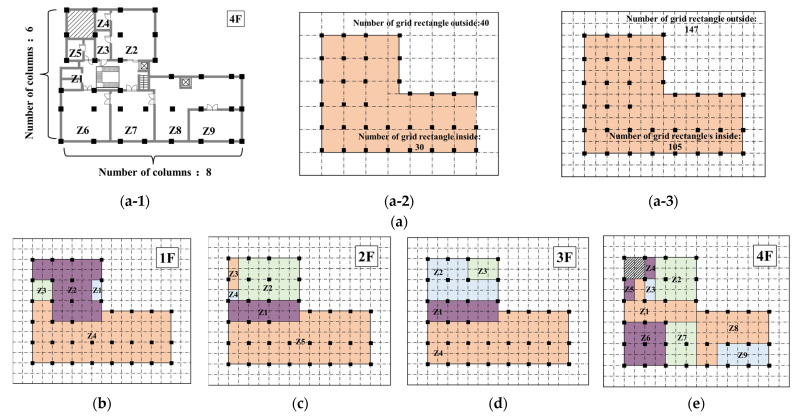
Grid formation process and results for the floor plan of target building A. (**a-1**) Number of columns along major sides. (**a-2**) Preliminary grid for the building floor plan. (**a-3**) Final grid for the building floor plan. (**a**) Example of grid formation for building floor plan for target building A. (**b**) Grid results on the 1st floor. (**c**) Grid results on the 2nd floor. (**d**) Grid results on the 3rd floor. (**e**) Grid results on the 4th floor.

**Figure 8 sensors-23-04116-f008:**
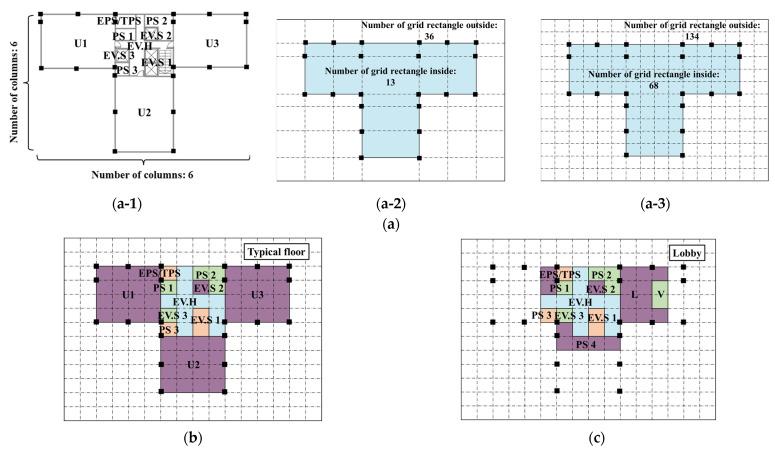
Grid formation process and results for the floor plan of target building B. (**a-1**) Number of columns along major sides. (**a-2**) Preliminary grid for the building floor plan. (**a-3**) Final grid for the building floor plan. (**a**) Example of grid formation for building floor plan for target building B. (**b**) Grid results for the typical floors. (**c**) Grid results on the lobby.

**Figure 9 sensors-23-04116-f009:**
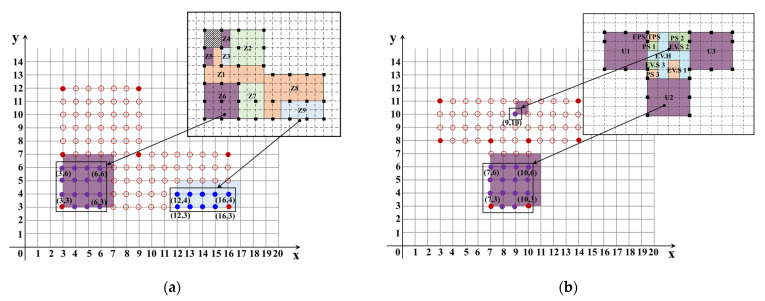
Examples of coordinate systems for the target buildings. (**a**) Coordinate system for measurement zones on the 4th floor of the target building A. (**b**) Coordinate system for measurement zones on the typical floor of the target building B.

**Figure 10 sensors-23-04116-f010:**
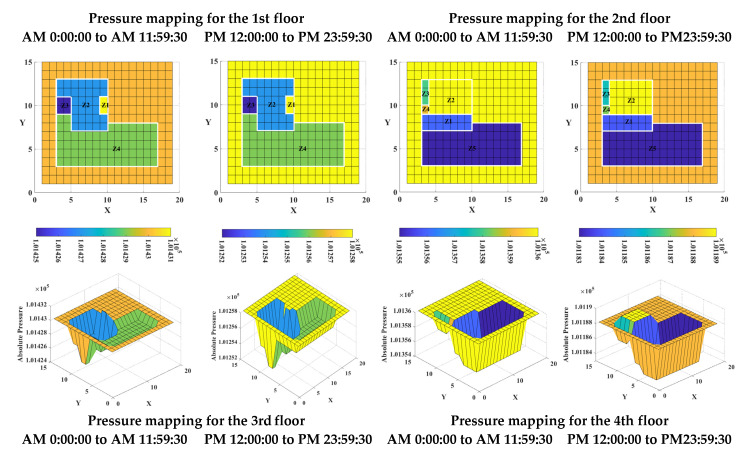
Pressure mappings for target building A.

**Figure 11 sensors-23-04116-f011:**
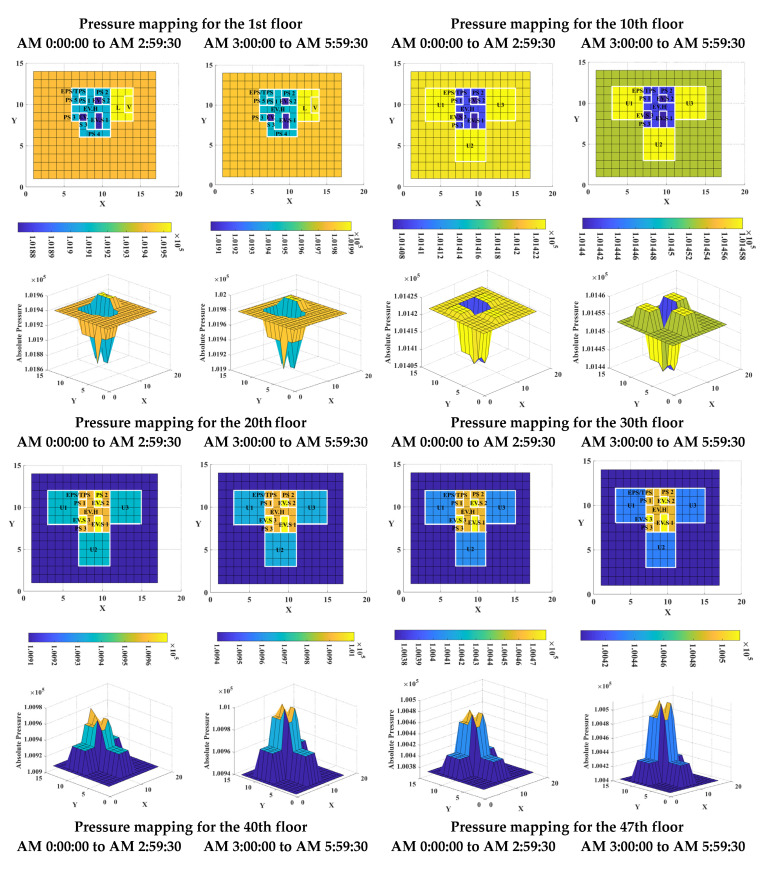
Pressure mappings for target building B.

**Table 1 sensors-23-04116-t001:** Specifications of pressure sensors equipped in the developed system.

Sensor Features	Specifications
Type	Barometer sensor: VAISALA PTB110
Operating range	Range: 800 to 1100 hPa
Accuracy	±0.30 hPa at 20 °C
Linearity	±0.25 hPa
Hysteresis	±0.03 hPa
Repeatability	±0.03 hPa
Interval	5 s (Minimum)
Storage	8 GB

**Table 2 sensors-23-04116-t002:** Summary of the target buildings.

Specification	Building A	Building B
Location	Incheon, Korea	Seoul, Korea
Completion (year)	2016	2020
Dimensions (m)	39.6 (L), 54.5 (W)	49.35 (L), 34.6 (W)
Height (m)	19.5	199.2
Number of floors (basement)	4 (0)	49 (5)
Usage	Office, Warehouse	Residence
Exterior	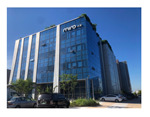	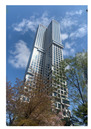

**Table 3 sensors-23-04116-t003:** Coordinate sets of the measurement zones for the target buildings.

Target Building	Zone	Coordinates Sets															
Building A (4th floor)	Z6	(3, 3),	(3, 4),	(3, 5),	(3, 6),	(4, 3),	(4, 4),	(4, 5),	(4, 6),	(5, 3),	(5, 4),	(5, 5),	(5, 6),	(6, 3),	(6, 4),	(6, 5),	(6,6)
	Z9	(12, 3),	(12, 4),	(13, 3),	(13, 4),	(14, 3),	(14, 4),	(15, 3)	(15, 4),	(16, 3),	(16, 4),						
Building B (Typical floor)	H2	(7, 3),	(7, 4),	(7, 5),	(7, 6),	(8, 3),	(8, 4),	(8, 5),	(8, 6),	(9, 3),	(9, 4),	(9, 5),	(9, 6),	(10, 3),	(10, 4),	(10, 5),	(10, 6)
	EV.S 2	(9, 10)															

**Table 4 sensors-23-04116-t004:** Mean pressure value of all zones during two measurement periods.

		**Mean Pressure (Pa)-AM 0:00:00—AM 11:59:30**					**Mean Pressure (Pa)-PM 12:00:00—PM 23:59:30**				
Building A	**Floor**	**Z1(Out)**	**Z2(Z6)**	**Z3 (Z7)**	**Z4(Z8)**	**Z5(Z9)**	**Z1(Out)**	**Z2(Z6)**	**Z3 (Z7)**	**Z4(Z8)**	**Z5(Z9)**
	1F	101,431	101,427	101,425	101,429	—	101,258	101,254	101,252	101,256	—
		101,430	—	—	—	—	101,258	—	—	—	—
	2F	101,356	101,360	101,358	101,359	101,355	101,184	101,189	101,186	101,187	101,183
		101,360	—	—	—	—	101,188	—	—	—	—
	3F	101,310	101,316	101,314	101,310	—	101,138	101,144	101,143	101,137	—
		101,313	—	—	—	—	101,141	—	—	—	—
	4F	101,265	101,266	101,259	101,262	101,261	101,094	101,095	101,088	101,090	101,089
		101,266	101,265	101,261	101,263	101,265	101,093	101,094	101,090	101,092	101,092
		**Mean pressure (Pa)-AM 0:00:00—AM 2:59:30**					**Mean pressure (Pa)-AM 3:00:00—AM 5:59:30**				
Building B	**Floor**	**EV.S**	**EV.H**	**L/U2**	**V**	**Out**	**EV.S**	**EV.H**	**L/U2**	**V**	**Out**
	1F	101,872	101,908	101,954	101,951	101,941	101,905	101,944	101,990	101,986	101,979
	10F	101,407	101,409	101,423	—	101,422	101,440	101,442	101,458	—	101,453
	20F	100,968	100,957	100,936	—	100,909	101,001	100,990	100,965	—	100,940
	30F	100,479	100,458	100,406	—	100,373	100,512	100,490	100,432	—	100,403
	40F	100,040	100,023	99,942	—	99,898	100,432	100,054	99,964	—	99,923
	47F	99,731	99,717	99,599	—	99,559	99,763	99,748	99,616	—	99,579

**Table 5 sensors-23-04116-t005:** Examples of spatial and numerical mapping relationships for the target buildings.

Target Building	Zone	Coordinates Sets					
Building A (4th floor)	Z6	(3, 3, 101, 265),	(3, 4, 101, 265),	(3, 5, 101, 265),	(3, 6, 101, 265),	(4, 3, 101, 265),	(4, 4, 101, 265),
		(4,5,101,265),	(4,6,101,265),	(5,3,101,265),	(5,4,101,265),	(5,5,101,265),	(5,6,101,265),
		(6, 3, 101, 265),	(6, 4, 101, 265),	(6, 5, 101, 265),	(6, 6, 101, 265)		
	Z9	(12, 3, 101, 265),	(12, 4, 101, 265),	(13, 3, 101, 265),	(13, 4, 101, 265),	(14, 3, 101, 265),	(14, 4, 101, 265),
		(15, 3, 101, 265),	(15, 4, 101, 265),	(16, 3, 101, 265),	(16, 4, 101, 265)		
Building B (20th floor)	U2	(7, 3, 100, 936),	(7, 4, 100, 936),	(7, 5, 100, 936),	(7, 6, 100, 936),	(8, 3, 100, 936),	(8, 4, 100, 936),
		(8, 5, 100, 936),	(8, 6, 100, 936),	(9, 3, 100, 936),	(9, 4, 100, 936),	(9, 5, 100, 936),	(9, 6, 100, 936),
		(10, 3, 100, 936),	(10, 4, 100, 936),	(10, 5, 100, 936),	(10, 6, 100, 936),		
	EV.S 2	(9,10,100,968)					

## Data Availability

Data available on request from the authors.

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
