# Peer review of "A Sensing-Based Visualization Method for Representing Pressure Distribution in a Multi-Zone Building by Floor"

_sensors, 2023, doi:10.3390/s23084116_

Round 1

Reviewer 1 Report

The research topic is interesting, and the results have the potential to enhance our understanding of pressure relations in buildings through sensing-based visualization. The manuscript is well written, with a clear objective, solid research methods, and good examples of pressure visualization in two buildings. However, the reviewer thinks that the benefits of pressure visualization are not adequately discussed in the manuscript. To strengthen the practical application discussion in the buildings, it is important to address the benefits of this new technology and potential cost issues. Additionally, as pressures in buildings are variable and change over time, it would be valuable to discuss how to obtain useful information from the continuously varying data obtained from pressure sensors. This discussion would be informative and could increase readership.

Reviewer 2 Report

The study focuses on an intuitive visualization method that   allows for the explanation of absolute pressure distribution in a multi-zone building using   accumulated data from a novel pressure-sensing system developed independently for pressure monitoring.

1. Objectives of the paper should be heighted in the introduction section.

2. At the end of the introduction section paper organization should added.

3. What is the strategy of deploying Slave devices?  Deployment of Slave devices should be optimal, is this thing considered while deployment?

4. What are external environmental parameters influencing the study?

5. Study of real time pressure monitoring is quite good and well explained.  Why such system is needed and what is the motivation behind this? Proper explanation is required, which will help reader to understand.

6.  The study is limited to low-rise and high-rise multi-zone buildings, if multiple buildings are together and gap among buildings are very less, what will happen in such scenarios?

7. thorough check is needed to correct spelling and grammar. (Like co-ordinate and coordinate, in-formation)

8. There too many papers from Build. Environ. journal included in the paper.

9. Flow diagram of whole methodology is required.

Round 2

Reviewer 2 Report

Well updated